# Role of Ape1 in Impaired DNA Repair Capacity in Battery Recycling Plant Workers Exposed to Lead

**DOI:** 10.3390/ijerph19137961

**Published:** 2022-06-29

**Authors:** Pablo Hernández-Franco, María Maldonado-Vega, José Víctor Calderón-Salinas, Emilio Rojas, Mahara Valverde

**Affiliations:** 1Departamento de Medicina Genómica y Toxicología Ambiental, Instituto de Investigaciones Biomédicas, Universidad Nacional Autónoma de México, Ciudad Universitaria, Mexico City 04510, Mexico; mastofsad@hotmail.com; 2Hospital Regional de Alta Especialidad del Bajío, Dirección de Planeación, Enseñanza e Investigación, Blvd. Milenio #130, Colonia San Carlos La Roncha, León 37660, Mexico; vega.maldonado.m@gmail.com; 3Departamento de Bioquímica, Centro de Investigación y Estudios Avanzados del Instituto Politécnico Nacional, Av. IPN #2508, Colonia San Pedro Zacatenco, Mexico City 07480, Mexico; jcalder@cinvestav.mx

**Keywords:** lead exposed workers, DNA-repair capacity, Ape1 activity, comet assay, stress and toxicity gene profile, ZF-TF

## Abstract

Exposure to lead in environmental and occupational settings continues to be a serious public health problem. At environmentally relevant doses, two mechanisms may underlie lead exposition-induced genotoxicity, disruption of the redox balance and an interference with DNA repair systems. The aim of the study was to evaluate the ability of lead exposition to induce impaired function of Ape1 and its impact on DNA repair capacity of workers chronically exposed to lead in a battery recycling plant. Our study included 53 participants, 37 lead exposed workers and 16 non-lead exposed workers. Lead intoxication was characterized by high blood lead concentration, high lipid peroxidation and low activity of delta-aminolevulinic acid dehydratase (δ-ALAD). Relevantly, we found a loss of DNA repair capacity related with down-regulation of a set of specific DNA repair genes, showing specifically, for the first time, the role of Ape1 down regulation at transcriptional and protein levels in workers exposed to lead. Additionally, using a functional assay we found an impaired function of Ape1 that correlates with high blood lead concentration and lipid peroxidation. Taken together, these data suggest that occupational exposure to lead could decrease DNA repair capacity, inhibiting the function of Ape1, as well other repair genes through the regulation of the ZF-transcription factor, promoting the genomic instability.

## 1. Introduction

Lead (Pb) is a heavy metal used by humans for many technological purposes. However, exposure to Pb is a reality, beyond the work environment. It has a natural origin, related to volcanic and geochemical activity, and impacts the environment causing water and soil contamination. Likewise, chronic exposure to low concentrations of lead is frequently caused by the consumption of food, traces of paint, forming part of particulate material that is inhaled, etc. Exposure to lead in environmental and occupational settings continues to be a serious public health problem [1,2,3]. At chronic exposure to high levels, Pb causes encephalopathy, kidney damage, anemia and toxicity to the reproductive system. Even at lower doses it may cause cognitive dysfunction, neurobehavioral disorders, neurological damage and hypertension [4,5,6]. In workers in the recycling lead industry, oxidative stress and erythrocytes apoptosis has been found [7,8,9].

Within the main mechanisms of action of metals, their ability to inhibit various DNA repair pathways has been considered [10,11,12]. It is important to keep in mind that it is through the optimal functionality of the repair mechanisms that genomic integrity is maintained, preventing cytotoxic and mutagenic effects of both exogenous and endogenous agents that damage DNA [13]. In fact, there is sufficient evidence of the risk of developing degenerative diseases, such as cancer, due to the inefficiency of any kind of genome safeguard mechanisms [14,15,16]. Hence the relevance of some reports, indicating that protein targets such as p53 and poly(ADPribose) polymerase 1 are inhibited by exposure to environmental heavy metals [17,18,19]. It was reported that exposure to Pb generates late and slowly repairable double-strand breaks that impact the mutated ataxia telangiectasia kinase–dependent stress signaling pathway by favoring error propagation and cancer proneness [11,20]. It is accepted that, within the mechanisms of greater weight for acquiring cancer or neurodegeneration, there is inhibition or dysfunction in the processes of DNA repair.

As one molecular target with respect to base excision repair, lead has been shown to inhibit the apurinic/apyrimidinic endonuclease (APE1) in low concentrations both in an isolated enzymatic test system and in AA8 cells [21] and Balb-c 3T3 fibroblasts [22], leading to an accumulation of apurinic sites in DNA and an increase in MMS-induced mutagenicity [23,24]. However, in exposed workers the molecular mechanism remains poorly understood.

The importance of continuing to report effects of mechanisms of action triggered by exposure to lead in humans lies in providing the International Agency for Research on Cancer (IARC) [25] with all possible data to have sufficient evidence to help demonstrate the possible carcinogenic effect of lead. Especially in light of the continuing occupational exposure to lead faced by adults in a variety of activities, from microelectronics and metallurgy, to building demolition, lead may be able to increase the risk of cancer by reducing the ability of the cell to repair DNA damage caused by other exposures rather than by causing alterations in DNA directly [26]. For these motives, the aim of the present study was to relate the limited repair capacity of DNA of human peripheral blood lymphocytes of workers exposed to lead in a battery recycling plant with the changes of expression in 112 stress and toxicity genes, and specifically with the expression of APE1 and its protein activity level.

## 2. Materials and Methods

### 2.1. Subjects

The study cohort was made up of all the staff of a battery recycling factory, who had a working period of between 3 to 6 years. The exposed group was composed of 37 workers (33 male and 4 female) that performed diverse activities and were occupationally exposed to lead. The non-exposed group was made up of 16 clinically healthy workers, 11 male and 5 female, without antecedents of occupational lead exposure. Inclusion criteria in both groups were did not present chronic-degenerative diseases or pharmacological treatment in the last 4 weeks prior to the beginning of this study. All subjects provided written, informed consent, and participation was voluntary. The Hospital Regional de Alta Especialidad del Bajío in México approved the study through the Investigation (CI/HRAEB/2018/057) and Ethics (CNBCEI-11-CEI-004-20170731; CEI-69-2018) Committees, in agreement with the Helsinki protocol.

### 2.2. Sample Collection

Venous blood was collected from each worker using heparin vacutainer tubes of five milliliters sealed Pb free (BectonDickinson, Indianapolis, IN, USA). Whole blood aliquots were stored at 4 °C until quantitation of lead concentration, delta-aminolevulinic acid dehydratase activity, lipid peroxidation, DNA damage and repair capacity. Subsequently, lymphocytes were isolated by the method of Ficoll-Hypake as previously described by Soto-Reyes et al., [27] to RNA, protein, and activity of APE1 test.

### 2.3. Blood Lead Concentration

The blood lead concentration was determined by voltammetry (Lead Analyzer, model 3010B, ESA Inc., Chelmsford, MA, USA) and reported as μg of Pb per dL of blood. All determinations were performed in duplicates. For this purpose, standard curves were used to minimize the effect of the matrix; it should be noted that the reported analyzes have a precision range of 87–104% (using ESA Hi and Lo calibrators) and the detection limit was 1 μg/dL [28]. It is important to mention that 10 μg/dL of Pb in blood is the limit permissible established by ATSDR [29]. 

### 2.4. Delta-Aminolevulinic Acid Dehydratase Activity (δ-ALAD)

δ-ALAD activity was measured in erythrocytes by the European standardized method as described by Berlin and Schaller [30]. The enzyme activity was determined spectro-photometrically (UV/VIS DU 650 (Beckman, Pasadena, CA, USA)) and expressed as nmol of porphobilinogen (PBG) per mL of erythrocytes per h (nmol/h/mL).

### 2.5. Lipid Peroxidation

The reactive species to thio-barbituric acid method (TBARS) was employed to analyze malondialdehyde (MDA) levels in erythrocytes. A 25 µL butylated hydroxy toluene (BHT) and 500 µL trichloroacetic acid (30%) aliquots was added to 500 µL of erythrocytes. After centrifugation at 1200 rpm, the supernatant was then added to EDTA and thio-barbituric acid; absorbency of the thio-barbituric acid-MDA complex was measured at 532 and 600 nm in a spectrophotometer (UV/VIS Beckman-DU650). The TBARS are expressed as nmol of malondialdehyde equivalents per mL of erythrocytes [31].

### 2.6. DNA Damage (Comet Assay)

From 5 µL of whole blood mixed with 75 µL of 0.5% low melting point (LMP) agarose, slide preparations were made. These preparations consisted of three layers; the first layer of previously dehydrated 0.5% normal melting point agarose, the second containing the blood cells and a third protection layer of 0.5% LMP agarose. The Comet assay was performed as closely as possible to that described in previous studies [32]. These preparations were subjected to cell lysis at 4 °C for at least 1 h (2.5 M NaCl, 100 mM EDTA, 10 mM Tris, pH 10, supplemented with 10% DMSO and 1% Triton X-100). Subsequently, they underwent the unwinding process in a horizontal electrophoresis chamber, immersed in an alkaline buffer solution (300 mM NaOH, 1 mM Na2EDTA, pH > 13) on a bed of ice for 20 min. At the end of this period, in the same electrophoresis chamber, a current of 300 mA and 25 V, ~0.8 V/cm for 20 min was applied to show single-strand breaks and alkali-labile sites. All technical steps were performed under indirect yellow light. As internal control, for every electrophoresis conducted, we introduce a slide containing human lymphocytes previously treated with 1 Gy of ionizing radiation. After electrophoresis, the preparations were subjected to pH neutralization by immersing them in neutralization buffer (0.4 M Tris, pH 7.5) at room temperature for 15 min with fresh buffer every 5 min. The preservation of the samples was achieved by dehydration with absolute ethanol and air drying. At the moment of evaluating the DNA damage, they were stained with 20 µL of an ethidium bromide solution (20 µg/mL) on the gel and with the help of an Olympus BX-60 microscope with fluorescence accessories (515-excitation filter) after which 560 nm, 590 nm barrier filter) and 20× magnification analysis was performed. Nucleoids analysis was performed with Komet v5.0, Kinetic imaging and 100 nucleoids per slide (duplicate) were evaluated per condition. 

### 2.7. Irradiation Procedure

Peripheral whole blood (30 μL) in 1 mL of PBS was exposed to 3 Gy of γ radiation in the Gammacell-1000 self-shielded irradiator with a sealed Cesium 137 source. Immediately after irradiation, the comet assay slides were made; meanwhile, the samples for recovery time were centrifuged at 5000 rpm for 15 s, the supernatant was discarded and fresh RPMI-1640 medium was added to incubate a 37 °C, 5% CO_2_ for the established time. After that, samples were centrifuged, discarding the supernatant, and the new comet assay slide prepared.

### 2.8. DNA Repair Capacity

A sound approach to measuring repair capacity is to inflict DNA damage on cells and monitor the persistence and/or removal of DNA-lesions induced over time. Therefore, cells were damaged with ionizing radiation (3 Gy) and the persistence or removal of breaks was monitored. After treatment, cells were centrifuged and cultured under optimal conditions, fresh culture medium, 37 °C, 5% CO_2_, and allowed to recover for 60 min. Subsequently, the Comet assay procedure was performed as just mentioned. Thus, we report the initial DNA damage of exposed and non-exposed workers to lead, we show the increased damage induced by the genotoxic challenge (3 Gy g radiation), and also the remaining damage at 60 min. For statistical analysis, a one-way ANOVA with Tukey HSD post hoc test was performed.

### 2.9. cDNA Expression Array

To determine gene expression profiles, SuperArray GEArray Q Series Stress and Toxicity Pathway Finder cDNA Array 1.2 nylon membranes (Clontech, Mountain View, CA, USA) were used. For this, total cellular RNA was extracted from isolated lymphocytes, which was hybridized with ^32^P-labeled cDNA probes, following the supplier’s protocol. The protocol used three cDNA probes from two independent RNA preparations for each sample, from both unexposed and lead-exposed workers. Hybridization was quantified from ^32^P label development using phosphorimager (Molecular Dynamics) and Atlas Image 1.0 software package (Clontech, Mountain View, CA, USA). Following is the list of the 112 genes that have the membrane, including blanks and housekeeping genes, numbered by position. ANXA5, ATM, BAX, BCL2L1, BCL2L2, CASP1, CASP10, CASP8, CAT, CCNC, CCND1, CCNG1, CDKN1A, CHEK2, CRYAB, CSF2, CYP1A1, CYP1B1, CYP2E1, CYP7A1, CYP7B1, DDB1, DDIT3, DNAJA1, DNAJB4, E2F1, EGR1, EPHX2, ERCC1, ERCC3, ERCC4, ERCC5, FMO1, FMO5, GADD45A, GADD45B, GPX1, GSR, GSTM3, HMOX1, HMOX2, HSF1, HSPH1, HSPA1A, PTGS1, HSPA1L, HSPA2, HSPA4, HSPA5, HSPA6, HSPA8, HSPA9B, HSPB1, HSPCA, HSPCB, HSPD1, HSPE1, IGFBP6, IL18, IL1A, IL1B, IL6, LTA, MDM2, MIF, PRDX1, PRDX2, MT2A, NFKB1, NFKBIA, NOS2A, PCNA, GDF15, POR, PTGS2, RAD23A, RAD50, CCL21, CCL3, CCL4, CXCL10, SERPINE1, SOD1, SOD2, TNF, TNFRSF1A, TNFSF10, FASLG, TP53, TRADD, UGT1A4, UNG, XRCC1, XRCC2, XRCC4, XRCC5, PUC18, PUC18, PUC18, Blank, Blank, Blank, GAPDH, GAPDH, PPIA, PPIA, PPIA, PPIA, RPL13A, RPL13A, ACTB, ACTB. Densitometric values were normalized to the cDNA for PPIA. The data was analyzed as follows: blank was subtracted from all values, additionally each value was normalized against the PPIA reference gene. The values of workers exposed to lead and those not exposed were averaged. Subsequently, the fold change was calculated as the ratio between the values of those exposed to lead and the values of those not exposed. At the value obtained from the ratio, its logarithm to base 2 was calculated. Genes that presented a log_2_ value greater than −0.86 with *p* < 0.05, which is 45% change, were considered down-regulated genes, while those genes that presented a log_2_ value greater than 0.53 with *p* < 0.05, which also is 45% of changes, were considered upregulated.

### 2.10. Reverse-Transcriptase-Polymerase Chain Reaction (RT-PCR)

RT-PCR reactions were performed using Access RT-PCR Systems (Promega, Madison, WI, USA), for which total RNA of isolated lymphocytes was extracted using Trizol (Invitrogen, Life Technologies, Carlsbad, CA, USA) according to the supplier’s instructions. Subsequently, the total RNA and its purity were quantified by spectrophotometry. Initial reaction conditions consisted of incubating samples at 42 °C for 60 min to prepare cDNA, 95 °C for 5 min to inactivate reverse transcriptase followed by 40 cycles of 1 min at 95 °C, 30 s at 60 °C and 1 min at 72 °C. °C It should be noted that GAPDH expression was used as a reference. Through the Primer express software (Applied Biosystems, San Francisco, CA, USA; ABI 2.0) the RT-PCR primers were designed, which were used and whose sequences were as follows:*APE1*-F-TAATTCTCTATCTCTGCCCC*APE1*-R-CAGTAATTCCCCGAAGCCTT*GAPDH*-F-AAACGACCCCTTCATTGACCT*GAPDH*-R-ATCTTAGTGGGGTCTCGCTC

### 2.11. Cell Protein Extractions

Isolated lymphocytes were suspended in phosphate buffered saline (NaCl 137 mM, KCl 2.7 mM, Na_2_HPO_4_ 10 mM and KH_2_PO_4_ 2 mM, pH 7.4) containing protease inhibitor cocktail (Calbiochem, Darmstadt, Germany) and lysate on ice by ten consecutive sonication rounds of 10 s with an ultrasonic homogenizer-4710 series (Cole-Parmer instruments, Vernon Hills, IL, USA) at power setting 40. The homogenate was centrifuged at 15,000× *g* for 10 min at 4 °C. Protein concentration was determined by Thermo scientific BCA protein assay kit and detected spectro-photometrically.

### 2.12. Western Blot for Ape1

Protein quantification of APE1 was performed by western blot [22]. Briefly, proteins were separated by 12% SDS-PAGE and transferred by electro-transfer to nitrocellulose membranes (Bio-Rad, Hercules, CA, USA) using standard transfer buffer. After transfer, blocking was performed in TBST (100 mM Tris-HCl, 150 mM NaCl, 0.1% Tween 20, pH 7.5) containing 2% skimmed milk powder for 60 min, washed three times. with TBST. Incubation with the primary anti-APE1 antibody (diluted 1:500) (Ref-1 (C4):SC17774, 37 KDa) was performed overnight. Three washes were then performed to continue the incubation for 60 min with the HRP-conjugated rabbit IgG secondary antibody diluted 1:20,000. Three washes with TBST were performed to reveal the membranes using the ECL detection system (Amersham International, Amersham, UK) according to the manufacturer’s recommendations. Bands were quantified by densitometric analysis and normalized to β-actin.

### 2.13. Ape1 Functional Assay

Ape1 activity was analyzed using a molecular beacon containing a synthetic tetra-hydrofuran (THF) residue, which mimics an AP site. (FITC)-d-(GCACTXAAGAATTCACGCCATGTCGAAATTCTTAAGTGC)-Dabcyl, where X is a THF Residue [33]. Briefly, a 20 μL aliquot of total cell extracts obtained from unexposed and lead-exposed workers in reaction buffer containing 20 mM HEPES (pH 7.5), 50 mM KCl, 2 mM EDTA, 1 mM β-mercapto-ethanol, and 0.1 mg/mL BSA. It is assumed that Ape1 will be in these extracts and will interact with the AP site contained in the molecular beacon in a period of 15 and 30 min at 37 °C, releasing the molecular beacon and therefore generating fluorescence. Said reactions were carried out in a flat bottom 96-well assay plate (Corning, Corning, NY, USA), with a final volume of 0.2 mL and fluorescence was measured with a FLx800 multiple detection microplate reader (BioTek, Highland Park, MI, USA) and they were analyzed with the attached software KcJunior v1.41.8 (BioTek, Highland Park, MI, USA). Excitation was at 488 nm and emission at 515 nm with fluorescence expressed as response units (RU).

### 2.14. Statistical Analysis

Statistical differences between non exposed and lead exposed workers for each of the determined parameters (lead concentration and δ-ALAD, lipo-peroxidation, using a Kruskall Wallis Two-way analysis. ANOVA, taking *p* < 0.05 as significant and Tukey HSP pos hoc test was employed to determine DNA repair capacity. For APE1 gene, protein expression, and molecular beacon studies, and for macro-array expression, a Mann Whitney *t*-test was performed. This was carried out using Prism 9.3.1, while for the correlations made between the measured parameters, Pearson’s simple correlation analysis was used with *p* < 0.05, using the Statistica program, version 6.0.

## 3. Results

### 3.1. Lead Exposure and Oxidative Stress Markers

Significant sources of occupational exposure to lead are the various processes involved in a lead-acid battery recycling plant. We studied a population chronically exposed by battery recycling. The epidemiological data and oxidative stress markers are summarized in Table 1. The mean blood lead concentration was 69.25 μg/dL in exposed group compared to 1.42 μg/dL in non-exposed group, showing a high occupational lead exposure in the recycling plant workers considering blood [Pb] permissibly limit of 10 ug/dL [29]. As oxidative stress markers, we evaluated the end products of lipid peroxidation, finding statistical difference in the quantification of malondialdehyde equivalents (MDA) in the exposed workers (1.52 nmol/mL) with respect the non-exposed workers (0.87 nmol/mL). Moreover, one of the most sensitive indicators of exposure and intoxication by lead, δ-aminolevulinic acid dehydratase (δ-ALAD) enzyme was strongly inhibited in the exposed workers (312.86 nmol/mL/h), compared with non-exposed (567.70 nmol/mL/h). These data indicate that the occupational lead exposure shows positive results in several oxidative stress markers.

### 3.2. Occupational Lead Exposure Decreases DNA Repair Capacity

Both DNA damage and reparative capacity were determined through the Comet assay, following the experimental design described in materials and methods. Our results show that non-exposed workers repair about 40% of the damage inflicted by radiation in the first 30 min, but after 60 min, all the damage inflicted on the DNA has been repaired, while exposed workers only repair 20% of the damage after 60 min. These results showed that DNA repair capacity is significantly lower in workers exposed to lead compared to the non-exposed group. DNA repair is visualized as persistence of DNA-SSBs after recovery under optimal conditions. This approach shows a similar basal damage between both groups of workers, the damage induced by the genotoxic challenge with 3 Gy of gamma radiation and the remnant damage after 60 min. It is possible to see that the group of exposed workers continue to show damage after 60 min, while the unexposed workers recover the basal level of damage (Figure 1).

### 3.3. Expression Profile of Toxicity and Stress Genes in Workers Exposed to Lead

There are several possible mechanisms by which lead could interfere with DNA repair capacity in exposed workers, one of which could be due to alterations in the expression of specific genes (Table 2). From the analysis of expression changes in 112 stress and toxicity genes, we found several upregulated genes with increases of at least 45% in lead exposed workers. Likewise, we found a down regulation of 28 genes in lymphocytes from lead exposed workers (Table 3).

Among the genes included in the Stress and Toxicity GE Array, there are genes for oxidative stress, heat shock, apoptosis, inflammation, DNA damage and repair. Of this array, only 31.25% showed changes with respect to the control. Of these, 80% were down regulated, while 20% of the genes were upregulated. These data reveal that chronic exposure to lead in these workers affects the expression of important genes in different signaling pathways, which together may be involved in the loss of DNA repair capacity among other cellular processes.

### 3.4. Impaired DNA Repair Capacity by APE1 in Lead Exposed Workers 

Considering the down regulation of BER genes and previous evidence from our group of APE1 inhibition, AP endonuclease of BER by in vitro exposure to lead [22], we evaluated its expression in exposed workers. APE1 was not included in the macro-array of Stress and Toxicity. Our results evidenced the decrease in APE1 mRNA expression by semi-quantitative PCR (RT-PCR). We show representative images of amplification products, comparing non-exposed and exposed workers (Figure 2A), as well as the densitometry quantitation of results for all subjects (Figure 2B).

The expression of mRNA does not necessarily correlate with the levels of protein expression; therefore, we quantified the expression of APE1 by western blot. From this trial, we also found a decrease in the protein level of Ape1 in workers exposed to lead. We show representative images of the gels (Figure 3A), as well as the graph of the densito-metric analysis of the percentage of expression of Ape1 (Figure 3B).

At this point, our results suggest that the loss of DNA repair capacity in exposed workers implies a decrease in APE1 expression at the gene and protein levels. Therefore, we consider it important to perform a protein functionality assay for Ape1. We choose a molecular beacon that is a DNA stem-loop structure that can be used to detect protein activity. A fluorophore and a quencher are attached to either end of the beacon. Without activity, the fluorescence is quenched but the structure unfolds in the presence of the Ape1 activity and fluorescence increase. 

In this case, we make use of a molecular beacon oligonucleotides containing an abasic site analogue (tetra-hydrofuran residue (THF) at position 5′ (FD-THF)) to measure Ape1 activity. We know that the modified molecular beacons with a short stem–loop region can interact in specific manner with Ape1 and are effective substrates for measuring DNA repair activities. As shown in Figure 4, incubation of FD-THF with cell-free extracts from lead-exposed workers reveals a decrease in fluorescence when compared with the average of fluorescent signal generated by FD-THF in cell-free extracts of non-exposed workers. Taken together, these data reveal for the first time that workers chronically exposed to lead displayed a transient inhibition of the incisions of the DNA due to a reduction in mRNA expression and protein levels of APE1 and impaired function of endonuclease activity of Ape1.

### 3.5. In Silico Prediction of Transcription Factor Binding Sites

Our data on stress and toxicity gene expression profile in lead exposed workers is a novel contribution. As mentioned, we found down-regulated expression levels of several DNA repair genes, and to explain their regulation pathway we carried out in silico analysis of the promoter sequences. We found several sequences of families of transcription factors present in the regulatory regions (Table 4). The result of this analysis shows eight well-represented families, being the family of transcription factors containing zinc fingers (ZF-TF) in which they are predicted to regulate a greater number of sub-expressed genes of the Stress and Toxicity GE-Array, as well as DNA repair genes.

### 3.6. Interaction of Oxidative Markers of Lead Exposure and the Loss of Ape1 Activity in Exposed Workers

Interactions and dependence of changes between markers of lead exposure and oxidative stress on loss of Ape1 activity in workers at a battery recycling plant were obtained by correlation analysis.

The results of Spearman’s correlation coefficients (r) between blood lead concentration (PbB), lead toxicity parameter (δ-ALAD activity), lipid peroxidation (MDA), Ape1 activity, APE1 gene expression (mRNA) and Ape1 protein expression among lead-exposed workers are presented in Table 5. A negative correlation coefficient was determined between (PbB) and δ-ALAD activity (r = −0.72; *p* < 0.001), Ape1 activity (r = −0.43; *p* < 0.01), APE1 mRNA level (r = −0.53; *p* < 0.001) and Ape1 protein level (r = −0.24; *p* < 0.01). The positive correlation coefficient was observed between (PbB) and (MDA) (r = 0.60; *p* < 0.001). δ-ALAD activity, which is a clinical lead toxicity parameter, had an inverse correlation with (MDA]) (r= −0.52; *p* < 0.001). In turn, δ-ALAD activity had a positive correlation with Ape1 activity (r = 0.38; *p* < 0.01) and APE1 mRNA level (r = 0.31; *p* < 0.01). Negative correlation coefficients were found between (MDA) and Ape1 activity (r = −0.33; *p* < 0.01) and APE1 mRNA level (r = −0.41; *p* < 0.001). The positive correlation coefficient was found between Ape1 activity and APE1 mRNA level (r = 0.37; *p* < 0.01). It is worth mentioning that the interaction between (PbB) and APE1 that we found through correlation analysis has already been reported in vitro by other studies. However, in this work we have found it in workers exposed to lead. This finding should be studied in greater detail to demonstrate that the physical interaction between APE1 and Pb is persistent in this type of occupational exposure, perhaps by chromatographic analysis, or DNA adducts. Elucidating this interaction will also clarify why we found a correlation between Ape1 activity and mRNA expression and not protein.

## 4. Discussion

The results described so far in the literature show that exposure to lead at concentrations above 10 μg/dL causes adverse effects on human health, including systemic dysfunction at very high concentrations [29]. Even at lower concentrations it can cause neurobehavioral disorders, neurological damage and hypertension [4,5,6]. In the present study carried out on workers at a battery recycling plant, we found that the average lead in blood was 69.25 ± 24.95 μg/dL, exceeding the permissible limit by more than six folds. We found in this group, oxidative markers related to lead intoxication, such as the inhibition of δ-ALAD activity and high concentrations of MDA as in many other works [30,31].

However, limited data is available regarding the impact of occupational lead exposure on DNA repair capacity [34,35]. Although there is evidence of the inhibition of the main DNA repair mechanisms by the effect of lead and other metals, this mechanistic and regulatory evidence has been considered from an experimental perspective [10,19,21,22,23,24,36]. In mammalian cells, an important mechanism for correcting DNA damage is the BER pathway. This is the most widely used pathway to deal with single base injury as well as to repair free radical-induced DNA single-strand breaks (SSB) [37], as DNA-lesions determined in the present study by Comet assay. Apurin apyrimidine endonuclease/redox effector factor 1 (APE1/Ref-1) plays a central role in the DNA BER pathway of DNA lesions such as uracil, alkylated and oxidized sites, and abasic, including single-strand breaks. It should also be considered that this endonuclease has transcriptional activity by modulating gene expression through ubiquitous transcription factors, such as AP-1, Egr-1, NF-kB, p53, HIF and tissue-specific (PEBP-2, Pax-5 and -8, TTF-1) [21,23,24,38]. For this reason, the relevance of this study lies in showing that in workers exposed to lead, there is a loss in DNA repair capacity related to a global profile of stress and toxicity genes, mainly involving genes of the BER mechanism, where APE1 plays a relevant role; however, other DNA repair pathways are also involved.

Lead can inhibit Ape1 activity in a concentration-dependent manner. This observation is consistent with biochemical studies performed on purified recombinant human APE1 and its bacterial exonuclease III homologue selectively inactivated by micro-molar concentrations of lead, and in whole cell extracts [21,23]. In agreement with these experiments, we found that, in the exposed workers from the battery recycling plant, there was a decrease in the gene and protein expression of Ape1, in addition to a loss in the functionality of the protein, which explains the loss in the reparative capacity of DNA lesions generated with ionizing radiation in lymphocytes of workers (Figure 2, Figure 3 and Figure 4 and Table 5). The inhibition of Ape1 activity may be due to the ability of lead to replace other polyvalent cations, as calcium and zinc, and impair various essential cellular functions or magnesium that is essential for the APE1 catalytic complex activity [24,37,39,40]. Lead competitively inhibits trace mineral absorption, binds to sulfhydryl proteins, disrupts calcium homeostasis, and reduces the level of available sulfhydryl antioxidant stores in the body, inhibiting DNA repair mechanisms [18,41]. Additionally, in the present work, the down regulation of a profile of 31 stress and toxicity genes is evidenced, with the BER, NER and HR mechanisms being more affected (Table 3). 

However, our data suggest that the inhibition of Ape1 activity in the lead exposed workers cannot be the only mechanism responsible for the decreased DNA repair capacity in these workers. Another possible mechanism for radiation-induced DNA repair inhibition is due to an altered gene expression of DNA repair genes. The results of Bae and collaborators [42] show decreased expression in *ERCC2, ERCC5, MSH2, TDG*, when exposing keratinocytes to metal mixture including lead. Moreover, treatment with different concentrations of lead acetate in PC12 cells decreased significantly the expression of *Bcl-2* and increased *p53* mRNA [43]. Changes in gene expression are often thought to be the indirect result of signal cascades, DNA methylation changes and ROS; however, metals may also be directly responsible for changes in transcription factor activity [44]. Evidence from this study show a decreased gene expression of DNA repair genes; *ERCC5, XRCC1, ERCC4, ERCC3, UNG,* and DDB1 in exposed workers versus non-exposed, using macro-array assay. Interestingly, workers lead intoxication results in a significant correlation between lead blood concentration and decreased *APE1* mRNA and protein levels, as well with the loss of APE1 functionality (Table 5).

Trying to understand the mechanisms underlying the changes in APE1 gene expression levels, we performed an in-silico analysis. Using promoter sequences of those genes that showed decrease in their expression, we observed that all the genes involved in DNA repair have more ZF-TF binding sites in their regulatory sequence (Table 4). Moreover, one mechanism through which lead and other metals may interfere with gene expression is by modulating the function and nature of transcription factors, which are responsible for gene activation or deactivation [12]. These experiments suggest that lead ions can alter the specific DNA binding ability of ZF-TF in vivo and in vitro [45]. All findings of the present study point out the importance of improving occupational medicine, to prevent any systemic acquisition of genome instability through dysfunction of BER, specifically APE1, that can lead to several diseases.

## 5. Conclusions

In summary, we found that chronic exposure to lead induced oxidative damage decreases DNA repair gene expression in recycling battery plant workers and can inhibit activity of APE1, which results in DNA repair capacity inhibition. More importantly, the present report for the first time provides evidence indicating that lead promotes increased DNA damage through inhibiting BER repair activity in workers exposed chronically to lead. The results suggest that reduced APE1 repair capacity by lead intoxication is a potential risk factor in environmentally related disease and plays an essential role in the geno-toxicity mechanism that is coupled to increased genomic instability, strengthening the notion that lead is a facilitator of carcinogenesis [22] and other non-communicative diseases.

## Figures and Tables

**Figure 1 ijerph-19-07961-f001:**
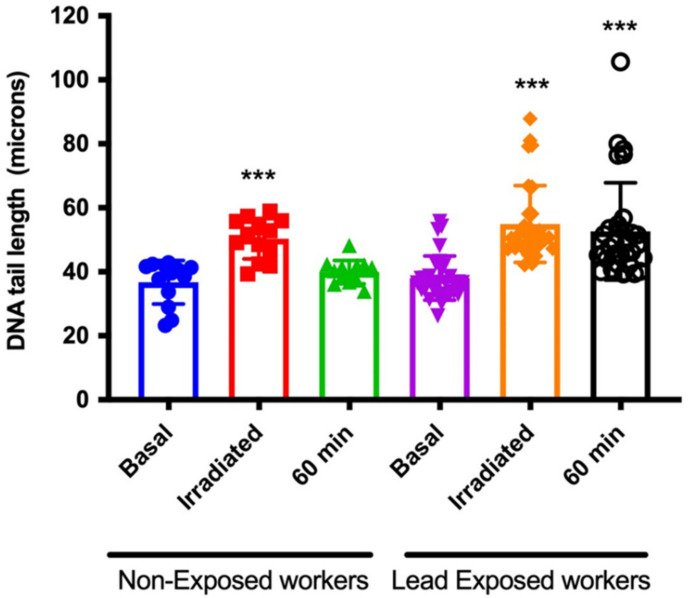
**Occupational lead exposure decreases DNA repair capacity**. Comet assay was performed from each individual (lead exposed and non-exposed workers), to show basal DNA damage; immediately after inflicting DNA damage with ionizing radiation (3Gy) and after 60 min, to determine DNA-repair capacity. Data are means ± S.E of 16 non-exposed and 37 lead exposed workers. One way ANOVA with Tukey HSD pos hoc test was performed with respect to the basal damage of both exposure groups. *** *p* < 0.0001.

**Figure 2 ijerph-19-07961-f002:**
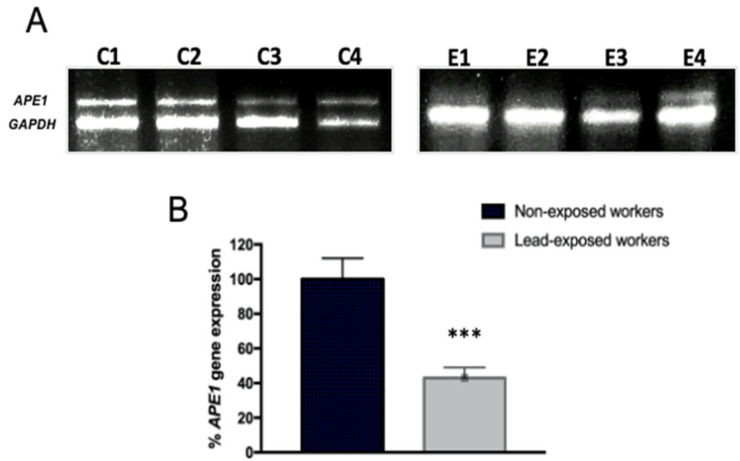
**Occupational lead exposure decreases APE1/Ref1 mRNA level.** mRNA extracts were prepared from lymphocytes of lead exposed and non-exposed workers and analyzed by semi-quantitative RT-PCR analysis of APE1/Ref1. (**A**) Representative gel image (C1–C4) of 16 non-exposed workers and representative image (E1–E4) of workers exposed to lead of all group of 37 workers. (**B**) Bar graphs showing relative densitometry quantification, data were normalized to GAPDH and represented as % as compared to control. Data are means ± S.E of three independent mRNA gels, of all samples. Mann–Whitney *t*-test *** *p* < 0.0001.

**Figure 3 ijerph-19-07961-f003:**
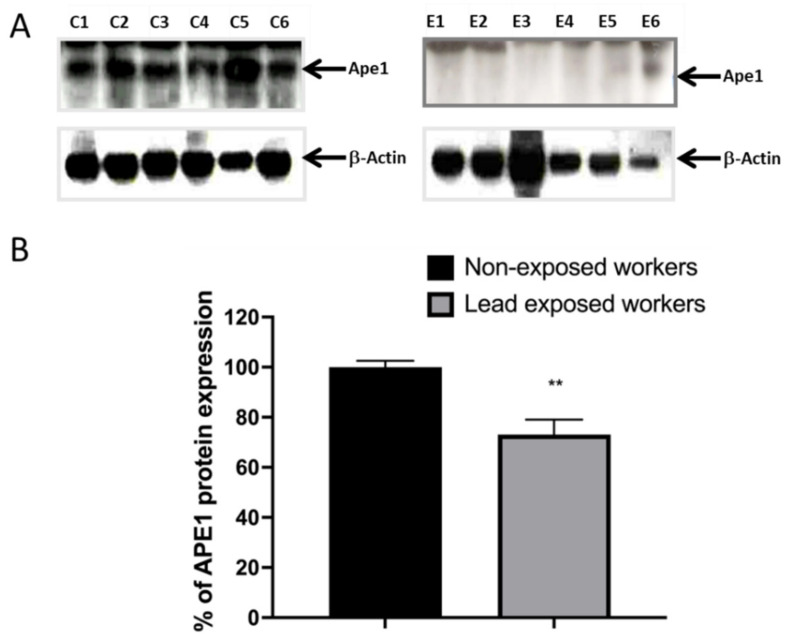
**Occupational lead exposure decreases the protein level of Ape1.** From total proteins extracted from lymphocytes of lead-exposed and non-exposed workers, APE1 immuno-blot was performed. (**A**) Representative immuno-blots (C1–C6) for Ape1 of all non-exposed workers (*n* = 16) and representative immuno-blots (E1–E6) of all lead exposed workers (*n* = 37). (**B**) Bar graphs show relative densitometry quantification, data was normalized to β-Actin and represented as % as compared to control. Data are means ± S.E of at least three independent western blots of all workers involved into the bio-monitoring. Mann–Whitney *t*-test. ** *p* < 0.01.

**Figure 4 ijerph-19-07961-f004:**
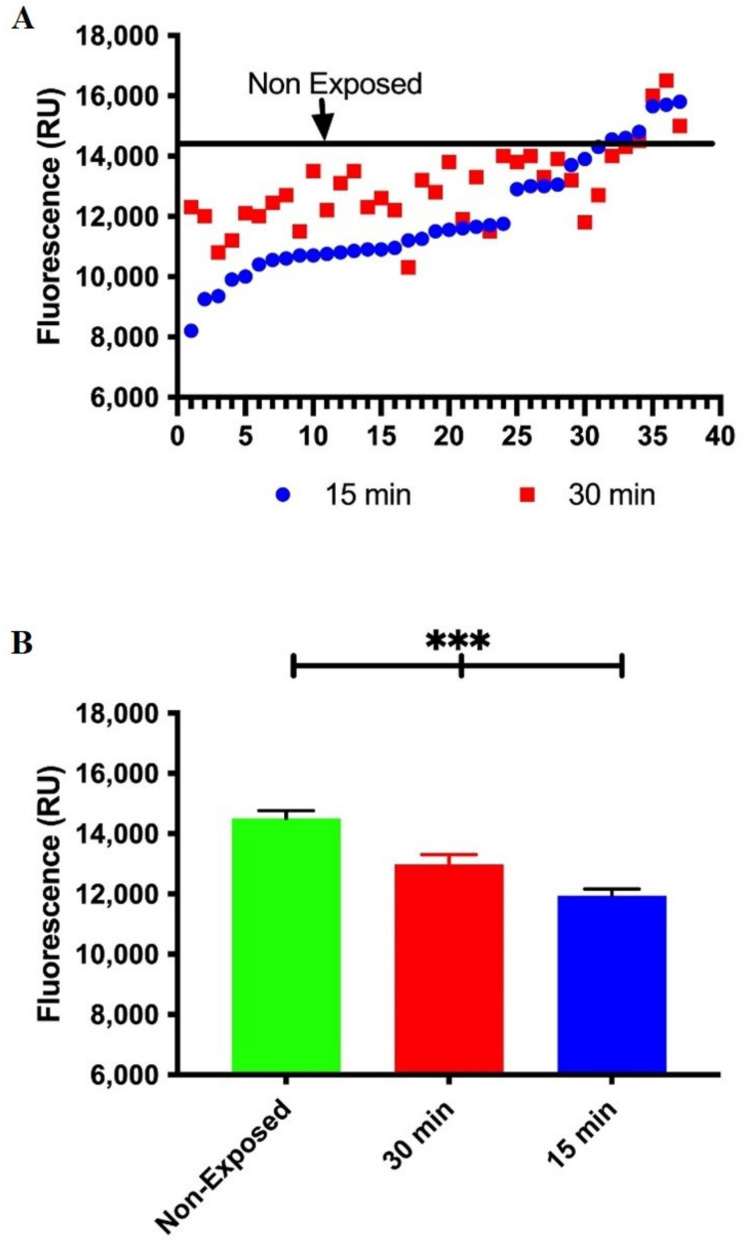
**Decreased Ape1 functionality in lead exposure workers.** Ape1 endonuclease catalytic activity was evaluated by molecular beacons-based assay in cell-free extracts from non-exposed and lead exposed workers. (**A**) Data in graph is represented as relative fluorescence units. Data are the fluorescence values of the reaction product of molecular beacon at 15 and 30 min of each lead-exposed worker. Line represents average relative fluorescence units of the non-exposed workers. Mann–Whitney *t*-test *** *p* < 0.0001. (**B**) Fluorescence quantification, represented as Mean ± SE of all unexposed workers, being those that show the highest Ape1 activity. We also show data for workers exposed to lead at 15 and 30 min.

**Table 1 ijerph-19-07961-t001:** Main characteristics of lead exposed and non-exposed workers (mean ± SD).

Variables	Non-Exposed (16)	Lead Exposed (37)
Age (years)	37.15 ± 7.09	31.65 ± 8.56
Duration of exposed (years)	-	4.53 ± 3.29
Smoking (%)	18.70	16.00
Blood lead concentration (µg/dL)	1.42 ± 0.87	69.25 ± 24.95 ***
ALAD activity (nmol/mL/h)	567.70 ± 46.20	312.86 ± 27.99 ***
MDA (nmol/mL)	0.87 ± 0.03	1.52 ± 0.08

*** *p* < 0.001.

**Table 2 ijerph-19-07961-t002:** Stress and Toxicity Gene expression changes in lead exposure workers and *p* values by *t* Test.

#	Gene	Exposed/NonExposed	FC Log_2_	*t* Test (*p* Value)	#	Gene	Exposed/Non Exposed	FC Log_2_	*t* Test (*p* Value)	#	Gene	Exposed/Non Exposed	FC Long_2_	*t* Test (*p* Value)
1	ANXAS	1.361	0.445	0.180	38	GSR	0.583	−0.778	0.001	75	PTGS2	1.347	0.429	0.128
2	ATM	0.850	−0.234	0.224	39	GSTM3	0.649	−0.623	0.005	76	RAD23A	1.307	0.386	0.146
3	BAX	1.374	0.459	0.132	40	HMOX1	0.546	−0.873	0.001	77	RAD50	0.883	−0.179	0.257
4	BCL2L1	1.046	0.064	0.413	41	HMOX2	1.807	0.853	0.072	78	CCL21	0.759	−0.397	0.073
5	BCL2L2	0.868	−0.204	0.254	42	HSF1	1.689	0.756	0.050	79	CCL3	0.799	−0.324	0.161
6	CASPS	0.836	−0.258	0.182	43	HSPH1	1.162	0.216	0.306	80	CCL4	0.609	−0.716	0.013
7	CASP10	0.862	−0.215	0.257	44	HSPA1A	1.074	0.103	0.395	81	CXCL10	1.728	0.789	0.012
8	CASP8	0.660	−0.599	0.036	45	PTGS1	0.657	−0.607	0.004	82	SERPINE1	1.565	0.646	0.041
9	CAT	1.752	0.809	0.071	46	HSPA1L	0.601	−0.734	0.000	83	SOD1	1.373	0.457	0.067
10	CCNC	1.521	0.605	0.095	47	HSPA2	0.684	−0.548	0.017	84	SOD2	0.910	−0.135	0.246
11	CCND1	1.280	0.356	0.210	48	HSPA4	0.582	−0.780	0.004	85	TNF	0.707	−0.500	0.011
12	CCNG1	0.902	−0.149	0.346	49	HSPA5	1.682	0.750	0.094	86	TNFRSF1A	0.807	−0.309	0.156
13	CDKN1A	0.728	−0.458	0.075	50	HSPA6	1.699	0.764	0.065	87	TNFSF10	0.718	−0.478	0.067
14	CHEK2	0.831	−0.268	0.166	51	HSPA8	1.150	0.202	0.297	88	FASLG	0.608	−0.718	0.011
15	CRYAB	1.297	0.375	0.157	52	HSPA9B	0.937	−0.093	0.411	89	TP53	1.332	0.413	0.079
16	CSF2	0.884	−0.178	0.307	53	HSPAB1	0.593	−0.753	0.000	90	TRADD	1.439	0.525	0.017
17	CYP1A1	1.495	0.580	0.120	54	HSPCA	0.577	−0.792	0.000	91	UGT1A4	0.841	−0.250	0.167
18	CYP1B1	1.638	0.712	0.086	55	HSPCB	0.709	−0.495	0.013	92	UNG	0.676	−0.565	0.007
19	CYP2E1	1.142	0.191	0.349	56	HSPD1	0.622	−0.684	0.003	93	XRCC1	0.638	−0.648	0.000
20	CYP7A1	0.837	−0.256	0.283	57	HSPE1	1.216	0.282	0.242	94	XRCC2	0.745	−0.424	0.073
21	CYP7B1	0.714	−0.486	0.052	58	IGFBP6	1.774	0.827	0.044	95	XRCC4	0.900	−0.151	0.312
22	DDB1	0.749	−0.417	0.059	59	IL1B	0.976	−0.036	0.462	96	XRCC5	0.788	−0.344	0.165
23	DDIT3	0.959	−0.061	0.430	60	IL1A	0.988	−0.017	0.483	97	PUC18	1.271	0.346	0.172
24	DNA1A1	0.678	−0.561	0.032	61	1L1B	0.631	−0.664	0.003	98	PUC18	1.232	0.301	0.148
25	DNAJB4	0.796	−0.328	0.163	62	1L6	0.712	−0.490	0.020	99	PUC18	0.920	−0.120	0.303
26	E2F1	1.431	0.517	0.118	63	LTA	0.739	−0.436	0.072	103	GAPDH	1.032	0.046	0.430
27	EGR1	1.015	0.022	0.478	64	MDM2	0.643	−0.636	0.020	104	GAPDH	0.950	−0.074	0.402
28	EPHX2	0.749	−0.417	0.175	65	MIF	1.607	0.684	0.072	109	RPL13A	0.837	−0.256	0.117
29	ERCC1	0.712	−0.490	0.080	66	PRDX1	1.928	0.947	0.020	110	RPL13A	0.875	−0.193	0.289
30	ERCC3	0.655	−0.611	0.008	67	PRDX2	1.314	0.394	0.078	111	ACTB	0.933	−0.101	0.383
31	ERCC4	0.640	−0.643	0.007	68	MT2A	1.101	0.138	0.030	112	ACTB	0.868	0.204	0.280
32	ERCC5	0.584	−0.777	0.004	69	NFKB1	0.770	−0.377	0.060					
33	FM01	1.274	0.350	0.210	70	NFKB1A	0.753	−0.409	0.054					
34	FM05	1.592	0.671	0.059	71	NOS2A	0.668	−0.583	0.025					
35	GADD45A	1.013	0.019	0.482	72	PCNA	0.591	−0.759	0.009					
36	GADD5B	1.053	0.074	0.427	73	GDF15	1.915	0.937	0.028					
37	GPX1	0.736	−0.441	0.036	74	POR	1.924	0.944	0.010					

Genes included in the Stress and Toxicity GE-Array, FC = fold change after normalization with respect to PPIA. # Genes are listed by array position; it is important to mention that APE1 in not included into the array.

**Table 3 ijerph-19-07961-t003:** Stress and Toxicity genes down or up regulated. Genes that presented a log_2_ value greater than −0.86 with *p* < 0.05, which is 45 % change, were considered down-regulated genes. While those genes that presented a log_2_ value greater than 0.53 with *p* < 0.5, which also is 45 % of changes, were considered upregulated.

Gene	Lead Exposed/Non Exposed	Fold Change log_2_	*t*-Test (*p* Value)	Up/Down
HSPCA	0.577	−0.792	6.8 × 10^−5^	Down
HSPA4	0.582	−0.780	0.004	Down
GSR	0.583	−0.778	0.001	Down
ERCC5	0.584	−0.777	0.004	Down
PCNA	0.591	−0.759	0.009	Down
HSPB1	0.593	−0.753	0.000	Down
HSPA1L	0.601	−0.734	0.000	Down
FASLG	0.608	−0.718	0.011	Down
CCL4	0.609	−0.716	0.013	Down
HSPD1	0.622	−0.684	0.003	Down
IL1B	0.631	−0.664	0.003	Down
XRCC1	0.638	−0.648	0.000	Down
ERCC4	0.640	−0.643	0.007	Down
MDM2	0.643	−0.636	0.020	Down
GSTM3	0.649	−0.623	0.005	Down
ERCC3	0.655	−0.611	0.008	Down
PTGS1	0.657	−0.607	0.004	Down
CASP8	0.660	−0.599	0.036	Down
NOS2A	0.668	−0.583	0.025	Down
UNG	0.676	−0.565	0.007	Down
DNAJA1	0.678	−0.561	0.032	Down
HSPA2	0.684	−0.548	0.017	Down
TNF	0.707	−0.500	0.011	Down
HSPCB	0.709	−0.495	0.013	Down
IL6	0.712	−0.490	0.020	Down
CYP7B1	0.714	−0.486	0.050	Down
GPX1	0.736	−0.441	0.036	Down
NFKBIA	0.753	−0.409	0.050	Down
SERPINE1	1.565	0.646	0.041	Up
HSF1	1.689	0.756	0.050	Up
CXCL10	1.728	0.789	0.012	Up
IGFBP6	1.774	0.827	0.044	Up
GDF15	1.915	0.937	0.028	Up
POR	1.924	0.944	0.010	Up
PRDX1	1.928	0.947	0.030	UP

**Table 4 ijerph-19-07961-t004:** In silico prediction of binding sites for different transcription factors.

Transcription Factors Families	Predicted Number of Downregulated Target Genes	Predicted Number of DNA Repair Genes
CEBP (Basic leucine zipper factors (bZIP))	16	6
SOX (High/mobility group (HMG) domain	16	13
HOX and POU (Home o domain factors)	38	24
FOS (Basic leucine zipper factors (bZIP))	20	17
MYC (Basic helix-loop/helix factors (bHLH))	21	18
E2F (Fork head/winged helix factors)	24	24
ELF/ELK (Tryptophan cluster factors)	26	22
Zn–TF (C2H2 zinc finger factors and nuclear receptors with C4 zinc finger)	140	72
TOTAL	1014	764

**Table 5 ijerph-19-07961-t005:** Pearson’s correlation coefficient (*r*) between oxidative markers of lead exposure and different Ape1 parameters in workers at a battery recycling plant.

	δ-ALAD Activity	[MDA]	Ape1 Activity	APE1 _m_RNA	Ape1 Protein
[PbB]	−0.72 ***	0.60 ***	−0.43 ***	−0.53 ***	−0.24 **
δ-ALAD activity		−0.52 ***	0.38 **	0.31 **	NS
[MDA]			−0.33 **	0.31 **	NS
Ape1 activity				0.37 **	NS
APE1 _m_RNA					NS

** *p* < 0.01, *** *p* < 0.001, NS Not significant.

## Data Availability

All data used is reported in the manuscript.

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
