# Peer review of "Role of Ape1 in Impaired DNA Repair Capacity in Battery Recycling Plant Workers Exposed to Lead"

_ijerph, 2022, doi:10.3390/ijerph19137961_

Round 1

Reviewer 1 Report

In the manuscript entitled “Role of APE1 in impaired DNA repair capacity in battery recycling plant workers exposed to lead” by Hernández-Franco and colleagues, the impact of lead exposure in the occupational setting was correlated with clinical parameters and the ex vivo DNA repair capacity, focussing on the major AP endonuclease APE1. While the findings are interesting and relevant for occupational health, there are some points that require clarification before publishing.

Major concerns:

  • Lines 75-83: Please provide more information about the cohort and applied statistics. Are all workers from the same factory? Only males? Furthermore, I could not find relevant information how the statistics was done to analyse the clinical data.
  • Line 85 and below: Make sure that is becomes clear whether you used whole blood extracts or just lymphocytes for the endpoint analyses, e.g. any kind of fractionation/enrichment done? E.g. RNA/protein level or activity in lymphocytes, excluding erythrocytes? Table 3,4, Figure 3,4???
  • Line 208: 69.25 µg/dL seems to be quite a lot of Pb. Would you call that level as severe intoxication? For the generally interested reader, it would be nice to find somewhere what the recommendations, limits, etc. are. Is there a correlation between the blood Pb levels and the duration of occupation?
  • Table 1: Obviously. smoking is a confounder. Why did you not do a multifactorial analysis here? A potential interaction between Pb and smoking would be interesting. And how was the statistic done?
  • Line 221 and Figure 1: Why did you not show the background DNA damage levels of the lead workers? The normalisation to time-point 0 after irradiation might be misleading as the damage induction/repair intermediates might also affected by Pb. Would it not be more meaningful to provide non-normalized data to assess the repair dynamics? Why was that done only for 3 samples out of 37/16 subjects? If so, how were they selected?
  • Line 234-249, Tables 2,3: Please provide the information about all genes analyzed. Was APE1 as key DNA repair gene not included in this panel? Does it contain the data from all probands? If not how many and why are there genes with p>0.05 (XRCC1, UNG) included? I do not think it makes any sense to do a GO term analysis here as the data is based on a panel of “stress genes”, which should inevitably return the selection criteria.
  • Figures 2, 3: You show the data 4 and 6 subjects but state in the Figure legends n=3. Was that not done for the whole cohort? The correlation done in Table 4 suggests so. Any selection made here? What is the rationale to make a non-parametric test on few samples, which is by the way not a t-test-based statistic? In Figure 3, there seems to be no band for APE1 in the Pb-exposed subject, differing with the quantification. Also the blots differ quite a bit compared to the controls and actin as standardizer seems to be highly saturated. How did you manage to quantify that?
  • Line 276-289, Figure 4: Why do you call that a “transient inhibition”? Could you color-code the data of exposed and non-exposed workers? This graph is not easy to understand.
  • Figure 5: Is this in silico analyses controlled against a random gene set or compared to the non-regulated genes in the panel? What is the meaning of the 1-10 kb sequences and how are they relative to genic features like the TSS. Needs to be better explained/controlled.
  • Table 4: Better describe the data size and statistics. How do you interpret the lack of correlation between APE1 protein levels and repair activity as well as mRNA levels? Is the presence of Pb in the extract directly inhibit APE1? That could be tested experimentally.

Minor concerns:

  • Overall, the manuscript is quite well written and easy to understand. Yet, some typos, odd phrasing and ambiguous wording could be improved. E.g.: lines 70-72, lines 217-218, line 252, line 335 (given is Pb in blood and not exposure level), inconsistencies in references,…
  • Line 28: the statement that SP1 is the key regulator in this context is speculative and would require some functional assessment
  • Line 45: reference 13 about genome engineering seems to be not the best one to make the point
  • Line 108: MDA per mL of erythrocytes, is that controlled for the erythrocyte concentration?
  • Line 123: specify the kind of ionizing radiation and the equipment used for exposure
  • Line 156-169: I doubt that this semi-quantitative approach is very reliable. Especially as the GapDH expression used as standardizer seems to better amplifying than APE1 and is more abundant (saturated bands!?, Figure 2) at 40 cycles, which is anyhow likely to be beyond the optimal reaction time. Could you not do the RT-qPCR by state-of-the-art means? At least, provide more information how the analysis and quantification is done.
  • Line 183: provide the name and provider of the antibodies
  • Line 214: there is no rating of the significance level. Either it is significant or not but there is no more or less

Author Response

Reviewer 1

Comments and Suggestions for Authors

In the manuscript entitled “Role of APE1 in impaired DNA repair capacity in battery recycling plant workers exposed to lead” by Hernández-Franco and colleagues, the impact of lead exposure in the occupational setting was correlated with clinical parameters and the ex vivo DNA repair capacity, focussing on the major AP endonuclease APE1. While the findings are interesting and relevant for occupational health, there are some points that require clarification before publishing.

Major concerns:

  • Lines 75-83: Please provide more information about the cohort and applied statistics. Are all workers from the same factory? Only males? Furthermore, I could not find relevant information how the statistics was done to analyse the clinical data.

A: We add more details of the study cohort in the Subjects section. Additionally, we introduce a section of Statistical analysis at the end of Materials and Methods.

  • Line 85 and below: Make sure that is becomes clear whether you used whole blood extracts or just lymphocytes for the endpoint analyses, e.g. any kind of fractionation/enrichment done? E.g. RNA/protein level or activity in lymphocytes, excluding erythrocytes? Table 3,4, Figure 3,4???

A: We has made this clarification in the sampling, corresponding material and methods sections and in the table and figure legends.

  • Line 208: 69.25 µg/dL seems to be quite a lot of Pb. Would you call that level as severe intoxication? For the generally interested reader, it would be nice to find somewhere what the recommendations, limits, etc. are. Is there a correlation between the blood Pb levels and the duration of occupation?

A: We introduce the permissibly limit of Pb into the paragraph and their reference. There is no correlation between Pb exposure level and duration of exposure.

  • Table 1: Obviously. smoking is a confounder. Why did you not do a multifactorial analysis here? A potential interaction between Pb and smoking would be interesting. And how was the statistic done?

A: Your observation is very pertinent. In this regard, through a bivariate analysis, we were unable to establish a relationship between smoking and blood lead exposure levels. Perhaps because the number of smokers in the group of exposed workers was low, only 6 of 37, which is the 16% of the exposed group (Table 1). In addition, in the group of unexposed workers, in which there are 3 smokers out of a total of 16, being 18% of smokers in non-exposed, in which the average blood lead levels were low (1.42mg/dL).

  • Line 221 and Figure 1: Why did you not show the background DNA damage levels of the lead workers? The normalisation to time-point 0 after irradiation might be misleading as the damage induction/repair intermediates might also affected by Pb. Would it not be more meaningful to provide non-normalized data to assess the repair dynamics? Why was that done only for 3 samples out of 37/16 subjects? If so, how were they selected?

A: We consider your comment pertinent, and we have changed Figure 1. In the current Figure 1, we show the kinetics of DNA damage, which allows us to visualize the difference in basal damage between the groups of workers and at the same time their sensitivity to radiation and its reparative capacity. At the same time, we show data corresponding to 60 minutes of recovery, due to is the major repair efficiency determined in both workers groups. We want to clarify that DNA-repair capacity was determined in all workers recluted in the biomonitoring. N=3, means the number of genotoxic challenges with 3Gy of radiation to assay repair capacity by Comet assay. We appreciate the suggestion since it strengthens the result.

  • Line 234-249, Tables 2,3: Please provide the information about all genes analyzed. Was APE1 as key DNA repair gene not included in this panel? Does it contain the data from all probands? If not how many and why are there genes with p>0.05 (XRCC1, UNG) included? I do not think it makes any sense to do a GO term analysis here as the data is based on a panel of “stress genes”, which should inevitably return the selection criteria.

A: You are right in the comment, therefore, we have included a more detailed description of the Array in the methodology. We include the list of the 112 genes contained in the array in which APE1 is not included. The data included in the tables correspond to the genes that responded to hybridization and that denoted an increase or decrease in their expression with respect to the non-exposed workers expressed as fold change in Log2 after housekeeping normalization. In such a way that we also carry out a new normalization analysis now with respect to GAPDH, because it was used to determine APE1 expression by mRNA and protein and thus have the same reference. It is right to point out that the terms of GO do not apply, however, we kept the classification in terms of the array used as a column into the new table 2. We change tables trying to clarify.

  • Figures 2, 3: You show the data 4 and 6 subjects but state in the Figure legends n=3. Was that not done for the whole cohort? The correlation done in Table 4 suggests so. Any selection made here? What is the rationale to make a non-parametric test on few samples, which is by the way not a t-test-based statistic? In Figure 3, there seems to be no band for APE1 in the Pb-exposed subject, differing with the quantification. Also the blots differ quite a bit compared to the controls and actin as standardizer seems to be highly saturated. How did you manage to quantify that?

A: Both figures 2 and 3 only show a representative image of an mRNA gel or western blot, respectively, but the desitometry data of all the workers is what is represented in sections "b" of both figures. It should be noted that the gels were made at least three times from the same samples; and that is why we had indicated N=3.

  • Line 276-289, Figure 4: Why do you call that a “transient inhibition”? Could you color-code the data of exposed and non-exposed workers? This graph is not easy to understand.

A: The entire paragraph was rewritten. We refer to a transient repair since it was evaluated at 15 and 30 minutes, observing changes. The suggested color code is included in Figure 4, making the difference in activity between workers exposed to lead at 15 and 30 minutes, since the control values are represented by the horizontal line indicated with an arrow.

  • Figure 5: Is this in silico analyses controlled against a random gene set or compared to the non-regulated genes in the panel? What is the meaning of the 1-10 kb sequences and how are they relative to genic features like the TSS. Needs to be better explained/controlled.

A: We realized a new in silico analysis and describe their principle. In the actual version, the results are presented as a Table 4.

  • Table 4: Better describe the data size and statistics. How do you interpret the lack of correlation between APE1 protein levels and repair activity as well as mRNA levels? Is the presence of Pb in the extract directly inhibit APE1? That could be tested experimentally.

A: We have added a statistics section in the methodology, and we have included data in the figure captions regarding the samples and the n.

On the other hand, we have included a paragraph in which we show our interpretation of the interaction between [PbB] and APE1 that we found in the correlation analysis, although this interaction has already been proved in vitro, by other works, in the present work, we have found it in exposed workers.

Minor concerns:

  • Overall, the manuscript is quite well written and easy to understand. Yet, some typos, odd phrasing and ambiguous wording could be improved. E.g.: lines 70-72, lines 217-218, line 252, line 335 (given is Pb in blood and not exposure level), inconsistencies in references,…

A: To lines 70-72 we made some improvements. Originally was "For these motives, the aim of the present study was investigating the ability of lead to induce modification in gene and protein expression of APE1, and its impact on the capacity of repair of DNA damage induced by ionizing radiation in human peripheral blood lymphocytes of workers who were chronically exposed in a battery recycling plant as well oxidative stress marker.". In the actual version: “For these reasons, the aim of the present study was to relate the limited repair capacity of the DNA of human peripheral blood lymphocytes of workers exposed to lead in a battery recycling plant with the changes of expression in 112 stress and toxicity genes, and specifically with the expression of APE1 and its protein activity level.”

To lines 217-218, the new paragraph is “These data indicate that occupational lead exposure shows positive results in several oxidative stress markers.”

We appreciate your observations, the writing for line 254 and 335 is changed. Thus, we verify the inconsistencies in references.

  • Line 28: the statement that SP1 is the key regulator in this context is speculative and would require some functional assessment
  • Line 45: reference 13 about genome engineering seems to be not the best one to make the point

A: We introduce new references.

  • Line 108: MDA per mL of erythrocytes, is that controlled for the erythrocyte concentration?

A: In effect, this was done according to the protocol reported in: Jain, S.K.; Ross, J.D.; Levy, G.J.; Duett, J, The effect of malonyldialdehyde on viscosity of normal and sickle red blood cells, Biochemical Medicine and Metabolic Biology, 1990, 44, 37-41, ISSN 0885-4505, doi.org/10.1016/0885-4505(90)90042 -Y. Therefore, this reference is included.

  • Line 123: specify the kind of ionizing radiation and the equipment used for exposure

A:

  • Line 156-169: I doubt that this semi-quantitative approach is very reliable. Especially as the GapDH expression used as standardizer seems to better amplifying than APE1 and is more abundant (saturated bands!?, Figure 2) at 40 cycles, which is anyhow likely to be beyond the optimal reaction time. Could you not do the RT-qPCR by state-of-the-art means? At least, provide more information how the analysis and quantification is done.

A: We understand your observation very well, we perform RT-PCR since it is the methodology to which we had access and we know its limitations. Of course, having had access to qRT-PCR, the quantifications would be more precise. However, the results obtained between the expression of mRNA and protein are consistent, in addition to being corroborated through an APE-1 activity test. It should also be noted that the quantifications shown in the graph correspond to the average of the determinations made for all the workers taking part in the study.

  • Line 183: provide the name and provider of the antibodies

A: Done

  • Line 214: there is no rating of the significance level. Either it is significant or not but there is no more or less

A: We change the paragraph.

Reviewer 2 Report

The manuscript by Hernández-Franco et al. describes toxic effects of occupational exposure to lead on DNA repair. They adequately characterize stress markers in the population under study and find less DNA repair capacity as based on the Comet assay, altered expression profile of genes related to DNA repair and decreased activity of the BER pathway endonuclease APE1. These findings had been reported before, at least partly. The conclusions sound reasonable, however, there are some inconsistencies, and the analysis is incompletely explained and/or not rigorously described. In particular, the language is not clear/correct enough, which makes discussion of the results seem poorly convincing. English needs to be carefully revised.

My concerns are the following:

Introduction: Lane 50: What do you mean by “asynchronous transfer mode”? Do you mean ATM (ataxia telangiectasia mutated) kinase...?

Materials and Methods:

Lane 105: “AFTER CENTRIFUGATION IN ... CONDITIONS...the supernatant was...

Lane 179: “The protocol for protein quantification of APE1 by means of western blot was the regulatory one used.” What do you mean by “the regulatory one”, the “regular/common one”? You should give a reference, especially since Western Blot is not a good method for protein quantification...

Results. Section 3.2 and Fig. 1:

I think that basal damage should be measured and taken into account: Might the repair machinery be less efficient in Pb-exposed cells because it is sequestered or engaged in massive DNA damage? In other words, is the initial (at t = 0 min, “100%”) damage really comparable? Which are the real initial values?

On the other hand, it has been reported that the alkaline conditions of the Comet assay may induce conversion of alkali-labile AP sites into SSBs, thus resulting in a possible overestimation of DNA damage (Miyamae Y. et al., 1997). This should be considered and discussed.

Fig. 1 legend: “Comet assay was performed IN BLOOD CELLS? from each individual..., immediately after INFLICTION OF...” “...and monitor the speed at which they remove the DNA-lesions at 30 and 60 minutes. I do not think that measuring at 30 and 60 min can be considered “a speed”. Rather, in a separate sentence, something like: “THE REMAINING DNA-LESIONS WERE MONITORED AT 30 AND 60 MIN”. You give the p-value meaning of asterisks, where in the figure are asterisks?

Section 3.3 and Table 2:

It is surprising that APE1 gene itself is not one of the differentially expressed ones in the comparative profile, especially considering the significant difference (2-3 folds) in mRNA levels described in the following section. This fact should be discussed / justified.

In Table 2, the entries should follow some order, either alphabetical (as they seem but do not obey) or rather, by fold change, in my opinion. The title for the “T-test” column should be “T-test p-value”...?

Section 3.4:

Fig. 2A: The two gel images should be more carefully aligned. Legend: “...from lymphocytes of lead exposed AND NON-EXPOSED workers ... AND ANALYZED by semi-quantitative RT-PCR analysis of APE1/Ref1. A) Representative gel imageS ... Bar GRAPH showING relative ... Data WERE normalized to GAPDH and represented as % AS COMPARED TO CONTROL”.

Fig. 3B: Should not the left bar correspond to exactly 100%? Legend: “Total PROTEIN WAS EXTRACTED from BLOOD SAMPLES? OF lead exposed AND NON-EXPOSED workers ...Data WERE normalized to ACTIN and represented as % AS COMPARED TO CONTROL”.

Lane 279: The “beacon assay” should be better explained, including the rationale of a fluorescence de-quenching. “We MAKE use of a ... oligonucleotide (singular) containing AN ABASIC SITE ANALOGUE (tetrahydrofuran...). The THF is NOT “at position 5´”. “... to MEASURE APE1 activity.” Lanes 281-282: I think that the beacon interacts specifically with APE1 (not “repair proteins” in general) not because of having a stem-loop structure but due to the THF residue. Lane 286: “...THESE data REVEAL for THE first time that workers chronically exposED to lead DISPLAYED (not “performed”) a transient inhibition of the DNA repair process (WHICH process, DNA incision...?)

Fig. 4: I guess that in this figure, data are ordered as a function of the value at t =15  min. Still, this should be explained, and X axis should be named. Here again, where are the asterisks? Importantly: if the activity at 15 min is higher than at 30 min, and it is stated that there is a “transient” inhibition, this should be more carefully analyzed with continuous kinetic records. Also, not only the individual data points but the average values with standard deviation should be reported. In addition, I find this result somehow contradictory with Fig.1: Is repair more inhibited at 15 min than at 30, and then again more inhibited at 60 min...?!

Section 3.5:

Lane 302: avoid the term “many”. Lane 303: “...specifically in THE BER pathway AS COMPARED to ... we show the motifs (WHICH ones, “other TF binding motifs”?) of transcription factor involved.” Involved in WHAT?

Fig. 5: The Y axis should be rather named something like: “Predicted number of TF-binding sites”..? The legend is also confusing, please rewrite.

Section 3.6:

Lane 316: Not “oxidative damage” but “LIPID PEROXIDATION”?

The correlation analysis should be briefly discussed / interpreted.

Table 4: Should be Table 3. What does “NS” mean, “not significant”?

Discussion:

Lane 340: “Although there is various (remove “various”) evidence of the inhibition of the main DNA repair mechanisms by the effect of lead and other metals; THIS mechanistic and regulatory evidence haS been considered from an experimental perspective” What do you mean “from an experimental perspective” I do not find anything negative in it.

Lane 359: Not “according” but “IN AGREEMENT WITH... we found that...”

Lane 364: “The inhibition of APE1 activity may be due to the ability of lead to replace other polyvalent cations, as calcium and zinc, and...[24, 36]” I do not think these references are adequate here. Might lead exert APE1 inhibition by substituting for magnesium, required for APE1 catalysis...?

Lane 368: “...reduces the level of available sulfhydryl antioxidant stores in the body, thereby inhibiting DNA repair mechanisms“. The term “thereby” is not adequate here, I do not see the cause – effect relationship.    

English should be corrected throughout the text, and many sentences are too long. I suggest some corrections only for the Introduction section:

Lane 38: “...even at lower doses IT (subject was lacking) may cause...”

Lane 46: “developing”

Lane 48: “... indicating THAT protein targets... polymerase 1 (remove “that”) are inhibited”

Lane 55: “has been SHOWN..”

Lane 58: “MMS-induceD”

Lane 60: “poorLY understood”

Lane 64: “Especially in light ...” This sentence is incomplete.

Lane 66: Remove “That is”

Lane 67: Remove “protect or”. “DNA damage CAUSED by...”

Lane 69: “... modificationS...”

Lane 70: Please split this very long sentence, and rewrite it: “The impact of lead on... as well as ITS VALUE AS? oxidative...”

Author Response

Answers to Reviewer 2.

Dear reviewer, we thank and appreciate your time to review the manuscript submitted for publication. We are sure that all your comments will contribute to improving the manuscript. We listened to each of your comments and observations, and we hope that they are reflected in the current version of the manuscript.

We kept the format you sent us and on top of it, we included our responses.

The manuscript by Hernández-Franco et al. describes toxic effects of occupational exposure to lead on DNA repair. They adequately characterize stress markers in the population under study and find less DNA repair capacity as based on the Comet assay, altered expression profile of genes related to DNA repair and decreased activity of the BER pathway endonuclease APE1. These findings had been reported before, at least partly. The conclusions sound reasonable, however, there are some inconsistencies, and the analysis is incompletely explained and/or not rigorously described. In particular, the language is not clear/correct enough, which makes discussion of the results seem poorly convincing. English needs to be carefully revised.

My concerns are the following:

Introduction: Lane 50: What do you mean by “asynchronous transfer mode”? Do you mean ATM (ataxia telangiectasia mutated) kinase...?

A: Yes, we think the spell checker introduced the "asynchronous transfer mode" bug, it should say mutated ataxia telangiectasia kinase, we attend the mistake.

Materials and Methods:

Lane 105: “AFTER CENTRIFUGATION IN ... CONDITIONS...the supernatant was...

A: Done, in the current version we indicate that in this step a centrifugation was performed and that is where the supernatant comes from.

Lane 179: “The protocol for protein quantification of APE1 by means of western blot was the regulatory one used.” What do you mean by “the regulatory one”, the “regular/common one”? You should give a reference, especially since Western Blot is not a good method for protein quantification...

A: We change the phrase and include a reference.

Results. Section 3.2 and Fig. 1:

I think that basal damage should be measured and taken into account: Might the repair machinery be less efficient in Pb-exposed cells because it is sequestered or engaged in massive DNA damage? In other words, is the initial (at t = 0 min, “100%”) damage really comparable? Which are the real initial values?

A: As you indicate, the repair capacity of DNA can behave differently in workers exposed to lead, therefore, we made a new Figure 1, in which we show the damage in basal DNA, that induced by the genotoxic challenge of gamma radiation and damage remaining after 60 minutes. This new representation has no adjustment, and it is possible to appreciate the inhibition of the reparative capacity of the peripheral blood lymphocytes of the workers exposed to lead.

On the other hand, it has been reported that the alkaline conditions of the Comet assay may induce conversion of alkali-labile AP sites into SSBs, thus resulting in a possible overestimation of DNA damage (Miyamae Y. et al., 1997). This should be considered and discussed.

A: We believed that in this case we are not overestimate the damage, due that one of the possible mechanisms of lead damage is precisely due to induction of AP sites but could be interesting to perform an analysis of the comet assay at pH 12.1, and evaluate how much DNA damage is due to AP-sites.

Fig. 1 legend: “Comet assay was performed IN BLOOD CELLS? from each individual..., immediately after INFLICTION OF...” “...and monitor the speed at which they remove the DNA-lesions at 30 and 60 minutes. I do not think that measuring at 30 and 60 min can be considered “a speed”. Rather, in a separate sentence, something like: “THE REMAINING DNA-LESIONS WERE MONITORED AT 30 AND 60 MIN”. You give the p-value meaning of asterisks, where in the figure are asterisks?

A: Thank you, we are agreed, 30 and 60 minutes cannot be considered as speed. We change presentation of this data and improve the figure legend.

Section 3.3 and Table 2:

It is surprising that APE1 gene itself is not one of the differentially expressed ones in the comparative profile, especially considering the significant difference (2-3 folds) in mRNA levels described in the following section. This fact should be discussed / justified.

A: The reason APE-1 is not included in the gene list in the tables is because it is not included in the GE-Array. Therefore, we determined its expression at the mRNA and protein levels by alternative methods.

In Table 2, the entries should follow some order, either alphabetical (as they seem but do not obey) or rather, by fold change, in my opinion. The title for the “T-test” column should be “T-test p-value”...?

A: We perform a new analysis and show new tables. One shows the entire array of genes included in the array (Table 2) and their expression change values, while Table 3 is sorting the data by significant fold change values for downregulated and upregulated genes.

Section 3.4:

Fig. 2A: The two gel images should be more carefully aligned. Legend: “...from lymphocytes of lead exposed AND NON-EXPOSED workers ... AND ANALYZED by semi-quantitative RT-PCR analysis of APE1/Ref1. A) Representative gel imageS ... Bar GRAPH showING relative ... Data WERE normalized to GAPDH and represented as % AS COMPARED TO CONTROL”.

A: Done

Fig. 3B: Should not the left bar correspond to exactly 100%? Legend: “Total PROTEIN WAS EXTRACTED from BLOOD SAMPLES? OF lead exposed AND NON-EXPOSED workers ...Data WERE normalized to ACTIN and represented as % AS COMPARED TO CONTROL”.

A: We already indicated the total protein were obtained from lymphocytes of lead exposed and non-exposed workers, and we adjust the left bar to 100%

Lane 279: The “beacon assay” should be better explained, including the rationale of a fluorescence de-quenching. “We MAKE use of a ... oligonucleotide (singular) containing AN ABASIC SITE ANALOGUE (tetrahydrofuran...). The THF is NOT “at position 5´”. “... to MEASURE APE1 activity.” Lanes 281-282: I think that the beacon interacts specifically with APE1 (not “repair proteins” in general) not because of having a stem-loop structure but due to the THF residue. Lane 286: “...THESE data REVEAL for THE first time that workers chronically exposED to lead DISPLAYED (not “performed”) a transient inhibition of the DNA repair process (WHICH process, DNA incision...?)

A: We rewrite the paragraph to explain the principle of the Molecular Beacon assay and also correct the location of the tetrahydrofuran residue.

Fig. 4: I guess that in this figure, data are ordered as a function of the value at t =15  min. Still, this should be explained, and X axis should be named. Here again, where are the asterisks? Importantly: if the activity at 15 min is higher than at 30 min, and it is stated that there is a “transient” inhibition, this should be more carefully analyzed with continuous kinetic records. Also, not only the individual data points but the average values with standard deviation should be reported. In addition, I find this result somehow contradictory with Fig.1: Is repair more inhibited at 15 min than at 30, and then again more inhibited at 60 min...?!

A: We insert a new graph with the average and the standard deviation, where it is clear than the activity at 15 minutes is less than at 30 minutes

Section 3.5:

Lane 302: avoid the term “many”. Lane 303: “...specifically in THE BER pathway AS COMPARED to ... we show the motifs (WHICH ones, “other TF binding motifs”?) of transcription factor involved.” Involved in WHAT?

A: This paragraph was removed.

Fig. 5: The Y axis should be rather named something like: “Predicted number of TF-binding sites”..? The legend is also confusing, please rewrite.

A: We perform a new in silico analysis and change the graphic by Table 4.

Section 3.6:

Lane 316: Not “oxidative damage” but “LIPID PEROXIDATION”?

A: Done

The correlation analysis should be briefly discussed / interpreted.

A: We introduce a paragraph.

Table 4: Should be Table 3. What does “NS” mean, “not significant”?

A: In the actual version this table correspond to Table 5, and include the legend NS- Not Significant.

Discussion:

Lane 340: “Although there is various (remove “various”) evidence of the inhibition of the main DNA repair mechanisms by the effect of lead and other metals; THIS mechanistic and regulatory evidence haS been considered from an experimental perspective” What do you mean “from an experimental perspective” I do not find anything negative in it.

A: Done

Lane 359: Not “according” but “IN AGREEMENT WITH... we found that...”

A: Done

Lane 364: “The inhibition of APE1 activity may be due to the ability of lead to replace other polyvalent cations, as calcium and zinc, and...[24, 36]” I do not think these references are adequate here. Might lead exert APE1 inhibition by substituting for magnesium, required for APE1 catalysis...?

A: We include a more accord reference.

Lane 368: “...reduces the level of available sulfhydryl antioxidant stores in the body, thereby inhibiting DNA repair mechanisms“. The term “thereby” is not adequate here, I do not see the cause – effect relationship.

A: Done    

English should be corrected throughout the text, and many sentences are too long. I suggest some corrections only for the Introduction section:

Lane 38: “...even at lower doses IT (subject was lacking) may cause...”

Lane 46: “developing”

Lane 48: “... indicating THAT protein targets... polymerase 1 (remove “that”) are inhibited”

Lane 55: “has been SHOWN..”

Lane 58: “MMS-induceD”

Lane 60: “poorLY understood”

Lane 64: “Especially in light ...” This sentence is incomplete.

Lane 66: Remove “That is”

Lane 67: Remove “protect or”. “DNA damage CAUSED by...”

Lane 69: “... modificationS...”

Lane 70: Please split this very long sentence, and rewrite it: “The impact of lead on... as well as ITS VALUE AS? oxidative...”

A: Done all of them.

Reviewer 3 Report

The article   “Role of APE1 in impaired DNA repair capacity in battery recycling plant workers exposed to lead” addresses an important topic in Public Health referred to gain more insight into the biological mechanisms involved in lead toxicity. The authors performed their investigation in blood samples obtained by human subjects volunteers occupationally exposed to lead as compared to non-exposed workers of a similar age range and smoking habits.

The manuscript is written in good English although some corrections should be addressed by the authors prior to publication.  The narrative of the research is clear and the presentation of the results is straightforward. The Material and Methods section should be improved (see comments below) with a description of the irradiation procedure and conditions. The discussion section is in accordance with their findings as well as the proposed interpretations of the data.

Comments and corrections

The grammatical correction suggestions are included in the form of a Comment balloon in the pdf of the manuscript.

Introduction

The authors clearly state the background knowledge and relevance of lead toxicity. Although they focused their research on occupational exposure, I suggest that the authors briefly address the present importance of lead as an environmental contaminant. Nowadays, lead contamination of water (drinking water and freshwater) is a very important issue of health concern.

Material and Methods

Since some biological responses are different in females and males, the number of subjects of each sex should be included in the description of the study.

One of the main points of this manuscript is the report of a decrease in the DNA repair capacity observed in irradiated blood samples of lead-exposed workers. However, there is no description of the irradiation procedure. Some of the information that should be added to the manuscript includes: the type of radiation used (ionizing radiation of different qualities could have been used eg. X-rays, gamma rays, accelerated electrons), the radiation source, dose rate, sample processing before, during and after radiation treatment, the time between radiation and start of Comet assay, how samples were irradiated (on ice, room temperature) and post-radiation treatment incubation. The authors should include a complete description of the radiation treatment.

Results

As shown in Fig 1 the percentage of DNA damage in irradiated lead-exposed samples at 30 min increased above the one observed at time 0. This is an interesting result that should be discussed by the authors given the upregulation of repair genes reported in Table 2.

Concerning the levels of APE1/ref1 mRNA and the protein levels, were these determined in the blood samples of the same subjects?

In summary:

Except for these comments and some additions required in the introduction and  methodology I have no other major comments about the study. Some suggestions to the authors will improve the manuscript. Therefore, in my opinion, this manuscript is suitable for publication if the comments and corrections are addressed in the text.

Author Response

Answers to Reviewer 3

Dear reviewer, we appreciate and thank you for your comments on the manuscript submitted for publication. We attended all of them and we are sure that this improves the manuscript.

We kept the format that you send us and on top of it we added the answers.

Comments and Suggestions for Authors

The article   “Role of APE1 in impaired DNA repair capacity in battery recycling plant workers exposed to lead” addresses an important topic in Public Health referred to gain more insight into the biological mechanisms involved in lead toxicity. The authors performed their investigation in blood samples obtained by human subject’s volunteers occupationally exposed to lead as compared to non-exposed workers of a similar age range and smoking habits.

The manuscript is written in good English although some corrections should be addressed by the authors prior to publication.  The narrative of the research is clear and the presentation of the results is straightforward. The Material and Methods section should be improved (see comments below) with a description of the irradiation procedure and conditions. The discussion section is in accordance with their findings as well as the proposed interpretations of the data.

Comments and corrections

The grammatical correction suggestions are included in the form of a Comment balloon in the pdf of the manuscript.

Introduction

The authors clearly state the background knowledge and relevance of lead toxicity. Although they focused their research on occupational exposure, I suggest that the authors briefly address the present importance of lead as an environmental contaminant. Nowadays, lead contamination of water (drinking water and freshwater) is a very important issue of health concern.

A: We include a paragraph at respect.

Material and Methods

Since some biological responses are different in females and males, the number of subjects of each sex should be included in the description of the study.

A: We include this data in the table 1.

One of the main points of this manuscript is the report of a decrease in the DNA repair capacity observed in irradiated blood samples of lead-exposed workers. However, there is no description of the irradiation procedure. Some of the information that should be added to the manuscript includes: the type of radiation used (ionizing radiation of different qualities could have been used eg. X-rays, gamma rays, accelerated electrons), the radiation source, dose rate, sample processing before, during and after radiation treatment, the time between radiation and start of Comet assay, how samples were irradiated (on ice, room temperature) and post-radiation treatment incubation. The authors should include a complete description of the radiation treatment.

A: We add a new section in material and method, explained the irradiation procedure.

Results

As shown in Fig 1 the percentage of DNA damage in irradiated lead-exposed samples at 30 min increased above the one observed at time 0. This is an interesting result that should be discussed by the authors given the upregulation of repair genes reported in Table 2.

A: A: Yes, it is an interesting result. For us, it could be due to the sum of the damage inflicted by g-radiation, which, when sensed by the repair mechanisms, generates incisions for the recruitment of the damage repair complexes of various DNA repair mechanisms, and therefore there is more damage than that generated after radiation. It also influences the fact that, through the panel of 112 stress and toxicity genes, we detected that exposure to lead generates a sub-regulation of several repair genes. However, the fact that at 60 minutes the damage decreases, indicates that the deterioration of the repair mechanisms is partial.

Concerning the levels of APE1/ref1 mRNA and the protein levels, were these determined in the blood samples of the same subjects?

A: Yes, all APE1 data were obtained all subjects participating for the study.

In summary:

Except for these comments and some additions required in the introduction and  methodology I have no other major comments about the study. Some suggestions to the authors will improve the manuscript. Therefore, in my opinion, this manuscript is suitable for publication if the comments and corrections are addressed in the text.

Round 2

Reviewer 1 Report

Dear authors

Having re-read the manuscript entitled “Role of APE1 in impaired DNA repair capacity in battery recycling plant workers exposed to lead” by Hernández-Franco and colleagues, I was pleased to see that the authors considered and addressed the majority of concerns raised by the reviewers. These substantial revisions are certainly in favour of the reader’s understanding of the data and conclusions but unfortunately also introduced some odds that need to be addressed prior to publication. I recommend that the authors sincerely proofread the manuscript once more, considering the non-exhaustive list of suggestions:

·         Abstract, line 34: Having removed the discussion about the SP1 TF, it should also be removed in the abstract

·         Odd phrases and typos: line 97, lines 110 ff, lines 169 ff, line 286 ff, line 346, line 382, line 394, line 513, …

·         Line 111: I did not find the description of lymphocyte purification by Ficoll gradients in the provided reference

·         Please, consistently use SI units with correct spacing

·         Figure 1, line 314: I am not sure whether 1-way ANOVA is the appropriate statistics here. These are 3 time-points for the same samples, aren’t they? Please, also provide the information about the post-hoc tests applied for pairwise comparisons

·         Line 326: none of the genes listed in Table 3 meets the mentioned criteria of 2-fold increase and statistical significance. And there are still genes with p>0.05 in the list. Please correct.

Author Response

Dear Reviewer 1,

Once again, we appreciate your time and comments in reviewing the submitted manuscript. Without a doubt, these final comments were of the utmost importance. We have listened to them and hope that the current version will be considered appropriate for publication.

Dear authors

Having re-read the manuscript entitled “Role of APE1 in impaired DNA repair capacity in battery recycling plant workers exposed to lead” by Hernández-Franco and colleagues, I was pleased to see that the authors considered and addressed the majority of concerns raised by the reviewers. These substantial revisions are certainly in favour of the reader’s understanding of the data and conclusions but unfortunately also introduced some odds that need to be addressed prior to publication. I recommend that the authors sincerely proofread the manuscript once more, considering the non-exhaustive list of suggestions:

  • Abstract, line 34: Having removed the discussion about the SP1 TF, it should also be removed in the abstract

A: You are completely right, we remove it.

  • Odd phrases and typos: line 97, lines 110 ff, lines 169 ff, line 286 ff, line 346, line 382, line 394, line 513, …

A: All of them were attended.

  • Line 111: I did not find the description of lymphocyte purification by Ficoll gradients in the provided reference

A: We made a mistake in the reference; we have included the correct one.

  • Please, consistently use SI units with correct spacing

A: We correct the SI units format across the manuscript.

  • Figure 1, line 314: I am not sure whether 1-way ANOVA is the appropriate statistics here. These are 3 time-points for the same samples, aren’t they? Please, also provide the information about the post-hoc tests applied for pairwise comparisons

A: You are right, we forgot including the post hoc test employed (Tukey HSD). Now it is included.

The one-way, or one-factor, ANOVA test for independent measures is designed to compare the means of three or more independent samples (treatments) simultaneously. In addition, we apply a Tukey test as pos hoc. The Tukey's HSD (Honestly Significant Difference) procedure facilitates pairwise comparisons within your ANOVA data. The F statistic tells you whether there is an overall difference between your sample means. Tukey's HSD test allows you to determine between which of the various pairs of means - if any of them - there is a significant difference.

  • Line 326: none of the genes listed in Table 3 meets the mentioned criteria of 2-fold increase and statistical significance. And there are still genes with p>0.05 in the list. Please correct.

A: We correct the Table 3.

Reviewer 2 Report

I think that the authors should have taken the opportunity to carefully revise their manuscript, not just inserting some of the corrections, and apparently they did not. In particular, the English is still too faulty and confusing. A reviewer cannot list 100% of the errors.

For instance, in the new version, in M&M, a section 2.8 is missing; there are different typing styles, "32" in "32P" should be superscript (lanes 193 and 196); "listing by position into the membrane" (lane 199) is a confusing expression; it should be UV/VIS instead of US/VIS in lane 126; "spectrophotometrically" instead of "spectrofotometrically", etc.

I think that Table 2 should be provided as Supplementary Material; other wise it is redundant with Table 3. In table 3, it should say p < 0.05, I guess. They should mention that APE1 is not included in the array; that is why they do alternative experiments...

Lane 342: This sentence is wrong, the complete paragraph should be rewritten.

The beacon assay is still not properly explained.

Table 4 is not properly explained: "target" of what? What do you mean by "sub-expressed" (not "sub-expresed")?

Lane 480: the name of APE1 is wrong, and it is the protein they study!

Author Response

Dear Reviewer 2,

We are sorry that you have that appreciation. We responded the comments including changes to the manuscript, we carried out new analyses, but unfortunately, the document obtained after accepting the changes to the revision made, caused serious errors.

We hope that the current version of the manuscript will change your opinion about the work.

I think that the authors should have taken the opportunity to carefully revise their manuscript, not just inserting some of the corrections, and apparently they did not. In particular, the English is still too faulty and confusing. A reviewer cannot list 100% of the errors.

For instance, in the new version, in M&M, a section 2.8 is missing; there are different typing styles, "32" in "32P" should be superscript (lanes 193 and 196); "listing by position into the membrane" (lane 199) is a confusing expression; it should be UV/VIS instead of US/VIS in lane 126; "spectrophotometrically" instead of "spectrofotometrically", etc.

A: We made all the changes.

I think that Table 2 should be provided as Supplementary Material; other wise it is redundant with Table 3. In table 3, it should say p < 0.05, I guess. They should mention that APE1 is not included in the array; that is why they do alternative experiments...

A: Done

Lane 342: This sentence is wrong, the complete paragraph should be rewritten.

A: The paragraph was rewritten.

The beacon assay is still not properly explained.

A: We rewrite all the section.

Table 4 is not properly explained: "target" of what? What do you mean by "sub-expressed" (not "sub-expresed")?

A: We change the terms.

Lane 480: the name of APE1 is wrong, and it is the protein they study!

A: Changed.